# Application Prospects of FTIR Spectroscopy and CLSM to Monitor the Drugs Interaction with Bacteria Cells Localized in Macrophages for Diagnosis and Treatment Control of Respiratory Diseases

**DOI:** 10.3390/diagnostics13040698

**Published:** 2023-02-12

**Authors:** Igor D. Zlotnikov, Alexander A. Ezhov, Maksim A. Vigovskiy, Olga A. Grigorieva, Uliana D. Dyachkova, Natalia G. Belogurova, Elena V. Kudryashova

**Affiliations:** 1Faculty of Chemistry, Lomonosov Moscow State University, Leninskie Gory, 1/3, 119991 Moscow, Russia; 2Faculty of Physics, Lomonosov Moscow State University, Leninskie Gory, 1/2, 119991 Moscow, Russia; 3Medical Research and Education Center, Institute for Regenerative Medicine, Lomonosov Moscow State University, 27/10, Lomonosovsky Ave., 119192 Moscow, Russia; 4Faculty of Medicine, Lomonosov Moscow State University, 27/1, Lomonosovsky Prosp., 119192 Moscow, Russia

**Keywords:** FTIR spectroscopy, CLSM, macrophage, latent infection, drug resistance

## Abstract

Visualization of the interaction of drugs with biological cells creates new approaches to improving the bioavailability, selectivity, and effectiveness of drugs. The use of CLSM and FTIR spectroscopy to study the interactions of antibacterial drugs with latent bacterial cells localized in macrophages create prospects to solve the problems of multidrug resistance (MDR) and severe cases. Here, the mechanism of rifampicin penetration into *E. coli* bacterial cells was studied by tracking the changes in the characteristic peaks of cell wall components and intracellular proteins. However, the effectiveness of the drug is determined not only by penetration, but also by efflux of the drugs molecules from the bacterial cells. Here, the efflux effect was studied and visualized using FTIR spectroscopy, as well as CLSM imaging. We have shown that because of efflux inhibition, eugenol acting as an adjuvant for rifampicin showed a significant (more than three times) increase in the antibiotic penetration and the maintenance of its intracellular concentration in *E. coli* (up to 72 h in a concentration of more than 2 μg/mL). In addition, optical methods have been applied to study the systems containing bacteria localized inside of macrophages (model of the latent form), where the availability of bacteria for antibiotics is reduced. Polyethylenimine grafted with cyclodextrin carrying trimannoside vector molecules was developed as a drug delivery system for macrophages. Such ligands were absorbed by CD206+ macrophages by 60–70% versus 10–15% for ligands with a non-specific galactose label. Owing to presence of ligands with trimannoside vectors, the increase in antibiotic concentration inside macrophages, and thus, its accumulation into dormant bacteria, is observed. In the future, the developed FTIR+CLSM techniques would be applicable for the diagnosis of bacterial infections and the adjustment of therapy strategies.

## 1. Introduction

Respiratory tract diseases (tuberculosis, pneumonia, mycoplasmosis etc) caused by pathogenic microorganisms are an acute problem in modern society [1,2]. Moreover, the resistant forms of microorganisms practically insensitive to antibiotics are caused by a number of factors, including the drug efflux effect, which is particularly dangerous [3,4,5,6,7,8]. For several decades, the main causative agents of bacterial forms of respiratory diseases have been *S. pneumoniae*, *M. tuberculosis*, *H. influenzae* type b, *S. pyogenes*, *M. catarrhalis*, and *S. aureus* [9]. These pathogens develop resistance to amoxicillin, rifampicin, macrolides, and cephalosporins. Resistance to antibiotics is multifactorial and may be caused by one or a combination of mutations in target genes, increased production of multidrug-resistant outflow pumps (MDR), or modifying enzymes and/or target-protecting proteins [10,11]. In this paper, special attention is focused on the study of efflux and ligands that increase specificity, which are considered as one of the main processes that reduces the effectiveness of antibiotics.

A special problem in the treatment of diseases are pathogens localized in macrophages and granulomas, thereby passing into a dormant form (but still dangerous) and are difficult to treat, as seen in some forms of tuberculosis, leishmaniasis, and respiratory diseases caused by chlamydial infections, which are particularly prevalent in children and people with weakened immunity, etc. [6,12,13]. This work is devoted to the development of spectral and optical methods for studying drug interactions with bacterial cells in their individual form and inside macrophages—as a model of latent infections. The considered approaches are perspective in application of the bacterial disease diagnosis by analysis of the FTIR spectra of pulmonary lavages using library data on microorganisms, as well as the optimization of the action of drugs for each patient. The possibility of analyzing pulmonary lavage is described in the work [14], where the combination of FTIR and confocal microscopy establishes the composition of the biological fluid [14]. However, for accurate validation, reference spectra and the use of several methods are necessary. The spectra biological objects such as cells are usually too multifactorial and it is not yet fully clear how to specify and isolate analytically significant signals; however, with the development of technologies, new prospects are opening up, such as time-resolved spectroscopy and 2D analysis. The literature describes the use of Raman scattering for the study of bacterial cells and identification of pathogens and strains [15]. Currently, the method is unreliable and, accordingly, not very informative, since it analyzes not the intrinsic spectra of substances in the giant Raman scattering, but only indirect ones—the effect of bacteria on the optical properties of nanoparticles. It strongly depends on the quality of the sample application, on the batch of particles, and on the conditions of strain cultivation [16,17].

On the contrary, FTIR spectroscopy, which is a highly informative method of analyzing chemical compounds, provides information about chemical bonds and the microenvironment of molecules, and is quite sensitive to changes in fine organization at the molecular level [6,18,19,20]. This makes it possible to study the structures of complex biological objects (cells, organelles), which is of limited use for other spectral methods that require the optical transparency of samples, which clearly does not apply to cells. Practical applications of FTIR spectroscopy include: analysis of biological tissues [21], tumor diagnosis [22], identification of pathogenic bacteria [23], and the study of molecular mechanisms of adaptation to changes in external conditions [24]—which opens up the opportunities to study the development of resistance. We assume that by using FTIR spectroscopy it would be possible to monitor the course of treatment, or the effectiveness of the drug used, as well as analyzing the biological fluids, regulating treatment regimes, and determining sensitivity to antibiotics. The advantages of FTIR spectroscopy include a small amount of substance for analysis (50 μL), non-invasive and giving numerical reagents, and biochemical changes at the molecular level. It has been widely discussed that the optical method of FTIR spectroscopy can potentially be used by doctors to accelerate the diagnosis of a patient or as an auxiliary method of analysis—during or before surgery [25].

Confocal Laser Scanning Microscopy (CLSM) is an optical imaging technique for increasing the optical resolution and contrast of micrography by using a spatial point hole to block out-of-focus light during image formation. Confocal microscopy allows direct, non-invasive sequential optical cutting of intact, thick, living samples with minimal sample preparation. Practical applications of CLSM include imaging of various tissues, cells, and drug interactions. CLSM provides efficient characterization of the physicochemical properties of drug delivery systems [26], diagnostics, and examination of cancerous tissues [27] and visualization of bacteria [6,28,29,30]. Confocal microscopy makes it possible to visualize the accumulation of drugs in cells and to study the mechanisms affecting the effectiveness of the therapeutic agent [6,28,29,31,32,33,34].

As possible promising experimental bases to solve the problem of latent infections localized in macrophages and multidrug resistance of bacteria, we propose two approaches: the use of a targeted drug delivery system to macrophages to concentrate drugs in the lungs [6,28,31,32,35,36,37,38,39,40,41,42,43,44,45,46,47,48,49,50,51,52,53,54,55,56], as well as the use of adjuvants (allylbenzenes and terpenoids) that inhibit efflux and increase the permeability of the membrane of pathogenic microorganisms [6,7,8,34,57,58,59,60,61]. We used rifampicin (Rif) [35,53,62] and doxorubicin (Dox) [35,57] as model (fluorophore) drugs, and studied their synergism with adjuvants (terpenoids from plant extracts) as well as the molecular mechanisms of such combined drug action that can be visualized by optical and spectral methods.

Thus, this study is aimed at developing a new approach based on spectral and optical methods for possible use in medical practice: monitoring the course of treatment based on spectral data of biological fluids, potential diagnosis of bacterial diseases, and strengthening existing therapeutic formulations.

## 2. Materials and Methods

### 2.1. Reagents

Carbonyldiimidazole (CDI) was obtained from GL Biochem Ltd. (Shanghai, China) via an intermediary Himprocess (Moscow, Russia). D-mannose, galactose, PEI 1.8 kDa (branched), fluorescein isothiocyanate (FITC), NaBH_3_CN, DMF, DMSO, Et_3_N, 2-hydroxypropyl-β-cyclodextrin (HPCD), 1M 2,4,6-trinitrobenzenesulfonic acid, rifampicin, and doxorubicin were obtained from Sigma Aldrich (St. Louis, MI, USA). Eosin-5-maleimide was purchased from Invitrogen (Molecular Probes, Eugene, Oregon, USA). Mannotriose-di-(N-acetyl-D-glucosamine) was obtained from Dayang Chem Co., Ltd. (Hangzhou, China). Eugenol and safrole at the highest commercial quality were purchased from Acros Organics (Flanders, Belgium). The preparation of apiol and plant extracts was carried out in the same way as described earlier [63]. Other chemicals such as salts and acids were obtained from Reakhim Production (Moscow, Russia).

### 2.2. Synthesis and Characterization of Conjugates 

#### 2.2.1. Synthesis of Grafted Chitosan and Cyclodextrin

The synthesis, purification, and characterization of conjugates were carried out as described earlier [6,20,50], including steps of HPCD activation by carbonyldiimidazole, subsequent conjugation with PEI amino groups, and modification by three types of carbohydrate labels: linear galactose, linear mannose, and trimannoside—to determine the affinity to CD206 receptors of macrophages. Introduction of the FITC label: to the aqueous solution of PEI1.8 (5% in 0.01M HCl, 1 g), a solution of FITC (15 mg in 1.5 mL DMSO) was added drop by drop with stirring; the pH was brought to 9.2 (sodium borate buffer, 0.1M). The mixture was incubated at 40 °C for 1 h, followed by purification by dialysis against water (cut-off 1 kDa) for 6 h. Purification by dialysis and HPLC: characterization was performed using by NMR, FTIR spectroscopy, analysis of nanoparticle trajectories, and dynamic light scattering.

#### 2.2.2. Dynamic Light Scattering (DLS)

The particle sizes and zeta potentials were measured using a Zetasizer Nano S «Malvern» (Worcestershire, UK) (4 mW He–Ne-laser, 633 nm, scattering angle 173°). The experiment was performed in a temperature-controlled cell at 25 °C. Autocorrelation functions of intensity fluctuations of light scattering were obtained using the correlation of the Correlator system K7032-09 «Malvern» (Worcestershire, UK). Experimental data were processed using «Zetasizer Software» (v. 8.02).

#### 2.2.3. Nanoparticle Tracking Analysis (NTA)

Determination of the hydrodynamic diameter of the synthesized polymers was carried out by NTA using a Nanosight LM10-HS device (Great Britain). Samples were diluted with MilliQ purified water to a concentration of 10^7^–10^9^ particles/mL and kept in an ultrasonic bath for 30 s. The hydrodynamic diameter was determined by the Stokes–Einstein equation relating to the analysis of the trajectory of the Brownian motion of particles. Each sample was measured three times. The hydrodynamic diameter of the particles was also determined using the method of dynamic light scattering.

### 2.3. Drug Loading

Loading of model fluorophores-antibiotics and adjuvants (eugenol) into HPCD-PEI1.8 delivery systems was carried out by 2-h incubation at 50 °C (0.005 M HCl)—a five–fold mass excess of polymer over the drug.

### 2.4. FTIR Spectroscopy Studying of the Antibiotic’s Actions on E. coli or CD206+ Macrophages Cells

*Escherichia coli* JM109 cells (overnight culture in liquid nutrient medium Luria–Bertani (pH 7.2), 10^8^ CFU) were washed twice with 0.01 M sterile PBS from the culture medium by centrifuging (Eppendorf centrifuge 5415C, 10 min, 12,000× *g*). Cell suspensions (10^7^ CFU/mL) were incubated with antibiotic samples; then, after 1-2-12-24 h, the cell’s samples were suspended and aliquots of 0.5 mL were taken. The cells are precipitated by centrifugation and separated from the supernatant, washed twice, and resuspended in 50 µL PBS to register the IR spectra. The supernatant is separated to determine the amounts of unabsorbed substances. ATR-FTIR spectra of cells samples suspension were recorded using a Bruker Tensor 27 spectrometer equipped with a liquid nitrogen-cooled MCT (mercury cadmium telluride) detector. Samples were placed in a thermostatic cell BioATR-II with a ZnSe ATR element (Bruker, Bremen, Germany). The FTIR spectrometer was purged with a constant flow of dry air (Jun-Air, Michigan, USA). FTIR spectra were acquired from 900 to 3000 cm^−1^ with 1 cm^−1^ spectral resolution. For each spectrum, 50–70 scans were accumulated at a 20 kHz scanning speed and averaged. Spectral data were processed using the Bruker software system Opus 8.2.28 (Bruker, Bremen, Germany), which includes linear blank subtraction, baseline correction, differentiation (second order, 9 smoothing points), min-max normalization, and atmosphere compensation. When necessary, 11-point Savitsky–Golay smoothing was used to remove noise. Peaks were identified by the standard Bruker picking-peak procedure. The concentration of Rif inside the cells was calculated from the material balance considering the unabsorbed Rif by UV-vis spectroscopy.

### 2.5. Macrophages Cell Lines

For the macrophage phagocytose assay, a human monocyte cell line THP-1 was used. Cells were obtained from the bank of cell lines at Lomonosov Moscow State University. THP-1 cells were cultured on T25 flasks in 5 mL RPMI-1640 (Gibco, Carlsbad, CA, USA), supplemented with GlutaMAX™ supplement (Gibco, Carlsbad, CA, USA) and buffered with 10 mM HEPES pH 7.4 containing 10% heat-inactivated FBS (Gibco, Carlsbad, CA, USA) and 1% antimycotic antibiotic (HyClone) at 37 °C and 5% CO_2_. To derive macrophage-like cells, THP-1 cells were seeded on 6-well plates in 2 mL of RPMI-1640 (Gibco, Carlsbad, CA, USA), supplemented with GlutaMAX™ supplement (Gibco, Carlsbad, CA, USA) and buffered with 10 mM HEPES pH 7.4 containing 10% heat-inactivated FBS (Gibco, Carlsbad, CA, USA) and 1% antimycotic antibiotic (HyClone) with the addition of 100 nM phorbol 12- myristate 13-acetate (PMA, p8139, Sigma Aldrich, St. Louis, MI, USA) for 72 h. After 72 h, the medium containing PMA was replaced with RPMI-1640 (composition described above) without PMA and cells were cultured for another 96 h.

CD206-evaluation. To block nonspecific binding sites, cells were incubated with a 10% solution of normal goat serum in PBS with 1% bovine serum albumin BSA for 1 h at RT. Then, the samples were incubated with a solution of anti-CD206 antibodies (ab64693, Abcam, 1:100) or rabbit polyclonal control IgG (910801, Biolegend) as a control for 2 h at RT and subsequently with goat-anti-rabbit antibody conjugated with Alexa594 (A11037, Invitrogen, 1:1000). The nuclei were labeled with DAPI (Sigma-Aldrich, St. Louis, MO, USA). Samples were analyzed with a Leica DM6000B fluorescent microscope equipped with a Leica DFC 360FX camera (Leica Microsystems GmbH, Wetzlar, Germany).

### 2.6. Confocal Laser Scanning Microscopy

*Escherichia coli* JM109 cells (overnight culture in liquid nutrient medium Luria–Bertani (pH 7.2), 10^8^ CFU) were centrifuged twice (Eppendorf centrifuge 5415C, 10 min, 12,000× *g*) and washed with 0.01 M PBS from the culture medium. Next, the cells were incubated for 60 min at 37 °C with 1 µg/mL of eosin-5-maleimide solution followed by twice-washing. Macrophages (CD206+ human monocyte cell line THP-1) placed in a 96-well fluorescent plate (Costar) were incubated for 1 h with eosin-labeled bacteria followed by washing (10 min, 4000× *g*). Samples (Dox in free form and with FITC-labeled delivery systems) were added to macrophages with absorbed *E. coli*. The cells were centrifuged twice with PBS washing (10 min, 4000× *g*). The cell centrifuge was suspended in 200 µL of PBS, followed by the addition 100 µL of a 5% agarose solution at 45 °C to solidify the cell suspension in the wells of a fluorescent plate. Fluorescence images were obtained by the confocal laser scanning microscope (CLSM) Olympus FluoView FV1000 equipped with both a spectral version scan unit with emission detectors and a transmitted light detector. CLSM is based on the motorized inverted microscope Olympus IX81. Emission fluorescence spectra of FITC (drug delivery system labelled), eosin (*E. coli* labelled), and Dox was obtained by CLSM. The excitation wavelength 488 nm (multiline Argon laser) and dry objective lens Olympus UPLSAPO 40X NA 0.90 were used for the measurements. Laser power, sampling speed, and averaging were the same for all image acquisitions. The scan area was 80 × 80 µm^2^. FITC, Eosin, and Dox fluorescence was collected using the emission windows set at 505–540, 540–575 nm, and 575–675, respectively, at 488 nm excitation. The signals were adjusted to the linear range of the detectors. Olympus FV10 ASW 1.7 software was used for acquisition of the images. FITC fluorescence is shown in green, Dox is red, Eosin is magenta, and the image on the light is gray.

### 2.7. Dox, FITC-Labelled Ligand, and Eosin-Labelled E. coli Determination Macrophage Uptake

Quantitative analysis of Dox, FITC-labelled ligand (as in paper [50]), and eosin-labelled *E. coli* (Section 2.6) content in CD206+ macrophages was performed using fluorescence spectroscopy. λ_exci_ (Dox or FITC) = 490 nm. λ_exci_ (eosin) = 515 nm. λ_emi_ (Dox) = 595 nm, λ_emi_ (eosin) = 560 nm, λ_emi_ (FITC) = 520 nm. Registration of fluorescence spectra was carried out using a SpectraMax M5 device (Pennsylvania, USA) in the Costar black/clear bottom tablet (96 wells). T = 25 °C. The concentration of Dox, FITC, and eosin inside the cells was calculated from the material balance considering the unabsorbed fluorophore’s concentration determined by fluorescence intensity. Intracellular concentrations of fluorophores were determined after destruction of macrophage cells by 10-min incubation with 1% Triton X-100 solutions.

### 2.8. Antibacterial Activity of Rif

The strain used in this study was *Escherichia coli* JM109 (J.Messing, USA). The culture was cultivated for 18–20 h at 37 °C to CFU ≈ 1.5 × 10^8^–2 × 10^8^ (colony-forming unit, determined by A600) in the liquid nutrient medium Luria–Bertani (pH 7.2) without stirring. The experiments in liquid media were conducted by adding 50 μL of the samples in 5000 μL of cell culture. The specimens were incubated at 37 °C for seven days. At the specific time, 100 μL of each sample was taken, diluted with distilled water, and the absorbance was measured at 600 nm. For quantitative analysis, the dependences of CFU (cell viability) on the concentration of Rif, 50 μL of each sample was diluted 10^5^–10^9^ times and seeded on the Petri dish. Dishes were placed in the incubator at 37 °C for 24 h. Then, the number of the colonies (CFU) was counted.

### 2.9. Statistical Analysis

A statistical analysis of the obtained data was carried out using the Student’s *t*-test Origin 2022 software (OriginLab Corporation, Northampton, MA, USA). Values are presented as the mean ± SD of three experiments (three replicates).

## 3. Results and Discussion

### 3.1. FTIR Spectroscopy of E. coli—Drug Interaction’s Tracking

FTIR spectroscopy can be effectively used to monitor the molecular details of the interactions of medicinal preparations with cells. In the cell, it is possible to distinguish the main structural units that contribute to the absorption of IR radiation (Figure 1): cell membrane lipids (2800–3000 cm^−1^), proteins, especially transmembrane (1500–1700 cm^−1^), DNA phosphate groups (1240 cm^−1^), and carbohydrates, including lipopolysaccharides (900–1100 cm^−1^). The main fluctuations of bonds in the structural units of E. coli cells were: 2960–2850 cm^−1^ CH, CH_2_, CH_3_ in fatty acids, 1655–1637 cm^−1^ amide I bands (α-helical and β-pleated sheet structures), 1548 cm^−1^ amide II band, 1515 cm^−1^ aromatic band, 1465–1470 C–H deformation, 1310–1240 cm^−1^ amide III band components of proteins, 1250–1220 and 1084–1088 cm^−1^ P=O stretching of PO_2_^−^ − phosphodiesters, and 1100–900 cm^−1^ C–O–C, C–O of saccharide ring vibrations [23]. The IR spectrum of lipids and phospholipids has the following characteristic peaks of functional groups: two bands of symmetric and asymmetric vibrations of hydrocarbon bonds, vibrations of the carbonyl group C=O, and vibrations of the phosphate (Figure 1). The position of the bands and their shape are sensitive to binding of the bilayer with ligands or drug molecules, hydrogen bond formation, aggregation, and oxidation, etc. [64].

To enhance the antibiotics efficiency, we used polymer nanoparticles HPCD-PEII1.8-triMan (polyethyleneimines grafted with cyclodextrins and with a carbohydrate labels on CD206 macrophage receptors (Table 1), as well as an adjuvant (on the example of eugenol), which inhibits the pumping of drugs from cells and increases bioavailability [6,50].

Rif and EG are only poorly soluble in water, so they need to be included in the delivery system, and the simplest is methyl-cyclodextrin (MCD). Further, the authors use cyclodextrin to prove the effectiveness of polymeric conjugates grafted by CD vs. simple cyclodextrin. Figure 2 shows the difference FTIR spectra (the spectrum is subtracted at zero time) of *E. coli* cell suspensions incubated with free Rif, Rif as part of a molecular container, EG in the form of an inclusion complex with β-cyclodextrin, and a combined formulation of antibiotic and adjuvant loaded into the delivery system. The aim of the experiment is to study the influence of the concentration of substances and the incubation time with cells on the changes in IR spectra of the cells, in other words, how the interaction of drugs and polymers with bacteria and macrophages is reflected on the spectra. The most pronounced changes are observed in the absorption bands of amides 1 and 2 (1600–1700 and 1500–1600 cm^−1^), oscillations of CH_2_ groups (2800–3000 cm^−1^), as well as in the region of 1240 cm^−1^ (PO_2_^−^ phospholipids and DNA) and 1000–1100 cm^−1^ (C-O-C carbohydrates).

Changes in the W1 region (Figure 1 and Figure 2) correspond to a change in the structural organization of the membrane and the accumulation of an antibiotic or EG inside the cells. For free Rif (Figure 2) in the first hour, there is a significant (*p*-value = 0.014) decrease in the intensity of ATR in W1, which indicates the incorporation of hydrophobic Rif molecules into the bacterial membrane (disordering of lipids). After 2 h of incubation, the antibiotic has already begun to accumulate inside the cells (the difference intensity in the IR spectrum decreases dramatically in the W1 region), which correlates with the data that the Rif penetrates into cells after 2–3 h (Figure 3). After 24 h, the antibiotic was eliminated by more than 70% cells (Figure 3), most probably because of efflux (quantitative determination of efflux was carried out by us earlier) [6]. The Rif concentration and ability to interact with transmembrane proteins correlate with the intensity of amide peaks 1 and 2 (Figure 2).

The inclusion of antibiotics in the delivery system to CD206+ macrophages (without EG) leads to significant (*p* < 0.003) effects on the accumulation of drugs inside cells (Figure 3). Accelerated drug absorption compared to free form is observed: (1) owing to the adsorption of polymer particles on the membrane surface (0.004–0.009 ATR vs. 0.001–0.004 in W2) and (2) owing to the occurrence of local defects in the membrane and increased penetration of the drug into the bacteria. In addition, the prolonged action of Rif in the delivery system is achieved, which can be observed by the changed in the intensity of amide 1 (Figure 2, top row). After 12–24 h, free Rif is characterized by a low intracellular concentration (Figure 3), and the drug in polymer particles is still working for 2–4 days [65,66]. The FTIR spectra of *E. coli* incubated with drug delivery system HPCD-PEI1.8-triMan itself are presented in Appendix A; the observed increase in intensity in the W2 region indicates the polymer carrier interacts with transmembrane proteins.

The interaction of EG–MCD (efflux inhibitor and enhancing membrane permeability agent [6]) with bacterial cells (Figure 2, bottom row) leads to: (i) inhibition of efflux pumps (as can be judged from increase in the intensity of amides 1 and 2) and (ii) the creation of defects (earlier was shown using CLSM [6]), which is reflected in a decrease W1 FTIR intensity and increase W3, corresponding to DNA and phospholipids). This explains the synergy of EG with antibiotics: we previously showed for levofloxacin and moxifloxacin enhancing antibacterial activity and we found enhanced absorption by CLSM in the sample with EG [6,20,50,63]. Similar effects are observed here for rifampicin by FTIR. The greatest effect is achieved for the combined antibiotic and adjuvant system in a polymer carrier (Figure 2, bottom row): strong amplification of W2 and W3 peaks, which correlates with blocking of efflux and high intracellular Rif concentration (Figure 3).

The changes discussed above in the FTIR spectra of *E. coli* when interacting with antibacterial agents reflect the accumulation of Rif inside cells (Figure 3). The concentration of free Rif does not exceed 1.5 μg/mL (calculated from the material balance considering the unabsorbed Rif by UV-VIS spectroscopy) and drops after 2–4 h of incubation. However, in the complex polymeric formulation, the penetration of the antibiotic is much more effective: a concentration of >2.5–3.5 μg/mL is achieved and, moreover, it is maintained at >2 μg/mL for 72 h.

### 3.2. FTIR Spectroscopy of E. coli in CD206+ Macrophages—Drug Interaction’s Monitoring

The CD206 mannose receptor is of greatest interest, which is involved in the recognition of pathogens stemming from the interaction of protein-binding domains with oligosaccharide patterns of microorganisms (*Candida albicans*, *Pneumocystis carinii*, *Leishmania donovani*, *Mycobacterium tuberculosis*, *Klebsiella pneumoniae*, etc.) [67,68]. The CD206 receptor mainly allows for targeting activated macrophages, in which resistant and dormant infections can accumulate. Selectivity toward micro-organisms is achieved because of the specificity of CD206 to mannose, fucose, and N-acetylglucosamine residues, which often cover the surface of pathogen cells, unlike mammals [31,41,49,56,69].

As shown above, the HPCD-PEI1.8-triMan molecular container and eugenol adjuvant enhance Rif penetration into bacterial cells and cause prolonged action according to FTIR spectroscopy data. Pathogens are least accessible when they localized in macrophages, so it is necessary to deliver antibacterial agents to macrophages, for example through the CD206 receptor.

Figure 4 shows the FTIR spectra of CD206+ macrophages with bacteria absorbed by them. We studied the interaction of macrophages with polymer carriers with three carbohydrate vectors of different affinity to CD206 (galactose—with low affinity to CD206, mannose with medium affinity and high affinity trimannoside vector), as well as the use of the EG adjuvant using FTIR spectroscopy, orthe effect of phagocytosis on the spectra. As shown earlier by flow cytometry [6,50] and confirmed here with FTIR spectroscopy (Figure 4), macrophages phagocytize polymer particles mainly with a high-affinity vector (triMan). Changes in the membrane of macrophages and E. coli are reflected in the region of 3000–2850 cm^−1^: the highest intensity means increased phagocytic activity and, consequently, greater accessibility for bacterial cells, which is further confirmed by an increase in the intensity of the peak of 1150–1000 cm^−1^ corresponding to the number of polymers adsorbed on *E. coli* and absorbed by macrophages. The CD206+ dependent binding of drug delivery systems to macrophages is confirmed by quenching peaks of amides 1 and 2, which is typical only for mannose-labeled polymers. Eugenol additionally enhances the accumulation of only high-affinity ligands (bottom row, Figure 4—amide region 1 and 2) and thereby increases the selectivity of the developed HPCD-PEI1.8-triMan carriers.

### 3.3. CLSM of E. coli in CD206+ Macrophages—Drug Interaction’s Visualization

To clarify the action mechanisms of polymeric carriers and adjuvant, CLSM and fluorescent studies of the drugs interaction with bacteria were carried out. We made a model system of macrophages with absorbed *E. coli*, which are colored with eosin, to study phagocytosis by macrophages of FITC-labeled HPCD-PEI1.8-triMan loaded with the fluorophore—antibiotic doxorubicin (Dox).

We studied three groups of samples:

(1) Control Dox to study the penetration and accumulation of free drug in macrophages and inside bacterial cells;

(2) Dox in a polymeric ligand (with different CD206-affinity labels: non-specific galactose, medium-affine mannose and high-affine triMan) to study macrophage phagocytosis activity and the effect of polymer on the adsorption efficiency on the bacterial cells and penetration of Dox;

(3) Dox in a polymeric ligand enhanced with EG as an agent enhancing the membrane permeability and efflux inhibitor.

The assignment of fluorescence signals (Dox, eosin, and FITC) is based on the fluorescence spectra of substances in the systems under consideration (Appendix A, Method section). Confocal images of macrophages with insider bacteria are shown in Figure 5 and Appendix A. On confocal images, large macrophages can be observed, inside of which *E. coli* are highlighted in pink, in which Dox accumulates asred dots.

Free Dox accumulates weakly in macrophages and inside *E. coli* (Figure 5a). Dox is released from the bacteria by pumping proteins in a process of efflux, and macrophages, in principle, poorly absorb small drug molecules.

The effect of adjuvant EG (Figure 5b) *on the accumulation of Dox.* An increase in the degree of Dox accumulation in bacterial cells in macrophages is observed. EG acts in two directions: creates defects in the membrane and inhibits pump proteins, as we showed in the last article [6].

The effect of drug delivery systems on the accumulation of Dox (Figure 5c–e). CD206-positive cells effectively phagocytosed predominantly high-affinity polymeric conjugate with trimannoside HPCD-PEI1.8-triMan, but not conjugate with linear galactose, which follows from the intensity of macrophage-associated fluorescence in the FITC channel (green). The label of linear mannose on the conjugate is medium effective. Inside the macrophages, colored dots are visible (in all channels), corresponding to the bacteria on which the polymer is adsorbed. Owing to the high penetration of HPCD-PEI1.8-triMan into macrophages, the accumulation of Dox in *E. coli* is very high, relative to control samples (Dox). Earlier, our cytometry assay determined that 80% of macrophage-like cells were FITC-positive after adding HPCD-PEI1.8-triMan, 60% were FITC-positive after adding HPCD-PEI1.8-Man, and 15% were FITC-positive after adding HPCD-PEI1.8-Gal [50]. Therefore, the data on CLSM and FTIR correlate with the flow cytometry data.

The synergy effect antibiotic and adjuvant, loaded into polymeric nanoparticles (Figure 5f and Appendix A). Dox in combination with eugenol, which enhances penetration through cell membranes and inhibits drug pumping, accumulates 10 times more efficiently than a simple substance in the composition of high-affinity conjugates to CD206+ macrophages. Thus, the complex formulation antibiotic and adjuvant in a drug delivery system is a perfect approach to accumulate the drug in the target macrophages with pathogenic bacteria.

In summary, by using CLSM and FTIR, we were able to distinguish bacteria inside macrophages, and showed how ligands penetrate into macrophages and thereby increase the accumulation of drugs inside bacteria.

Thus, confocal microscopy confirms the data of FTIR spectroscopy in the terms of enhancing the permeability of the bacterial membrane to the drug caused by polymers and adjuvants (efflux inhibitors) and phagocytosis by macrophages. Therefore, spectral changes are confirmed visually and quantitatively (Section 3.4. Table 2).

### 3.4. Quantitative Data on the Penetration of Drugs into Macrophages with E. coli

Table 2 presents quantitative data on the absorption of Dox and FITC-labeled conjugates (determined based on the material balance of extracellular and intracellular concentrations in macrophages after lysis with 1% Triton X-100, fluorescent detection). The data on ligand uptake correlate with those previously obtained by flow cytometry: the carrier with the trimannoside vector is absorbed by cells by more than 60–70%, and galactose-labeled by only 10–15%. Thus, owing to polymer ligand and adjuvants (eugenol or its analogues), it is possible to increase the accumulation of Dox inside macrophages by more than three times. Taking into account the previously obtained values for isolated *E. coli* [6], the accumulation of Dox directly in the bacteria increased by more than 10 times (it implies the total effect of enhanced penetration into macrophages x 3–4 and then in *E. coli* inside macrophages × 3).

### 3.5. Rif Antibacterial Activity on E. coli

Using FTIR spectroscopy, changes in cells during incubation with drugs have been demonstrated—therefore, it is important to show how these data correlate with antibacterial activity. The antibacterial activity of Rif in a free form and in the composition of molecular containers and enhanced with eugenol demonstrates correlations between the observed effects using spectral and optical methods and what is actually observed in a microbiological experiment. Figure 6 shows the curves of the survival of *E. coli* bacterial cells on the incubation time with various forms of Rif in comparison with the control (without the addition of an antibiotic). Apparently, polymers accelerate the penetration of antibiotics into cells because of the adsorption of polysaccharides on the cell wall surface. According to the data on CFU and the turbidity of the cell suspension (A600), the HPCD-PEI1.8-triMan polymer itself does not affect cell growth, and the Rif efficiency practically does not increase when incorporated into cyclodextrin (MCD); however, when loaded into the polymer system, the efficiency increases significantly (0.5–1 CFU order). Eugenol in the form of a complex with MCD demonstrates antibacterial activity; however, it is not bright in itself. At the same time, eugenol acts as a synergist as an adjuvant to the antibiotic Rif, so cell growth is quickly suspended, and the number of viable cells falls. Previously, the synergism of antibiotics (moxifloxacin and levofloxacin) with eugenol, menthol, and apiol was demonstrated by us [20,63,70]. Thus, the complex formulation of antibiotic and booster adjuvant in the delivery system with active targeting function is a promising combination for the possible strengthening of existing therapies.

Thus, the correlations of the antibacterial activity of therapeutic agents with the observed effects in the FTIR spectra and visually in a confocal microscope are demonstrated. Optical methods are applicable for the diagnosis and optimization of the treatment of bacterial infections.

## 4. Conclusions

Visualization of the action of therapeutic agents at the molecular level seems to be a powerful way to find out the points of increasing the effectiveness of antibacterial drugs, and in the future, cytostatic ones. The methods of FTIR spectroscopy and confocal microscopy are potentially applicable for the diagnosis of latent infections localized in macrophages, such as pneumonia, tuberculosis, and mycoplasma. Using FTIR spectroscopy, it was shown that the accumulation of rifampicin in model *E. coli* cells increases with the use of polymeric molecular containers and adjuvants that inhibit efflux and increase membrane permeability. The penetration of the model fluorophore antibiotic doxorubicin into CD206+ macrophages with *E. coli* localized in them was visualized using CLSM. The efficiency of phagocytosis of polymer drug carriers by macrophages depending on the carbohydrate label (galactose, mannose, trimannoside) was compared. The developed delivery systems increase the effectiveness of the therapeutic agent (Dox or Rif) by more than 10 times (it implies the total effect of enhanced penetration into macrophages ×3–4 and then inside macrophages in *E. coli* ×3). Thus, we presented the potential of practical application of optical and spectral methods (FTIR + CLSM) in aspects of drug study and possible diagnosis of diseases.

## Figures and Tables

**Figure 1 diagnostics-13-00698-f001:**
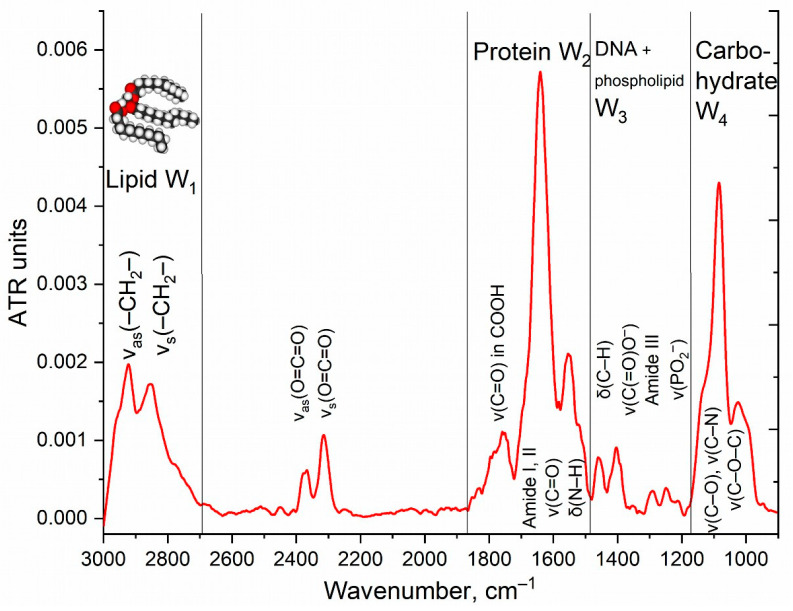
FTIR spectra of *E. coli* cells suspension in water. T = 22 °C.

**Figure 2 diagnostics-13-00698-f002:**
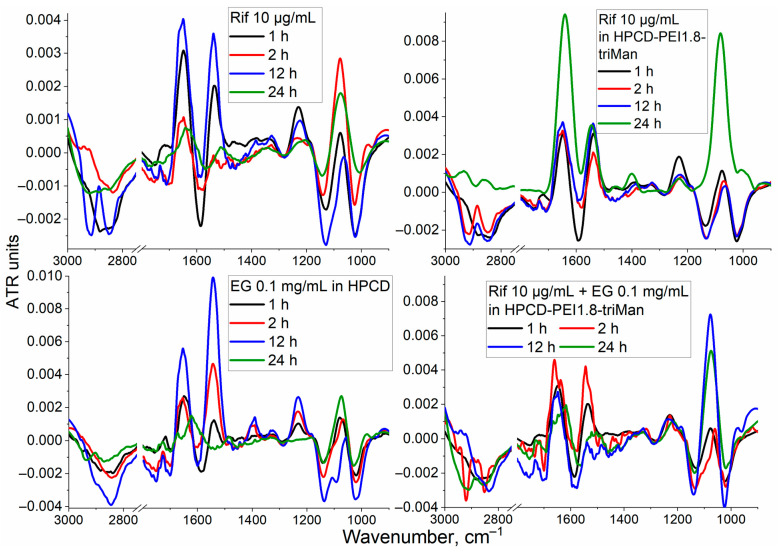
Difference FTIR spectra of *E. coli* incubated with rifampicin (Rif) or/and eugenol (EG) drug formulations. The spectra at the zero moment of time are subtracted. T = 22 °C.

**Figure 3 diagnostics-13-00698-f003:**
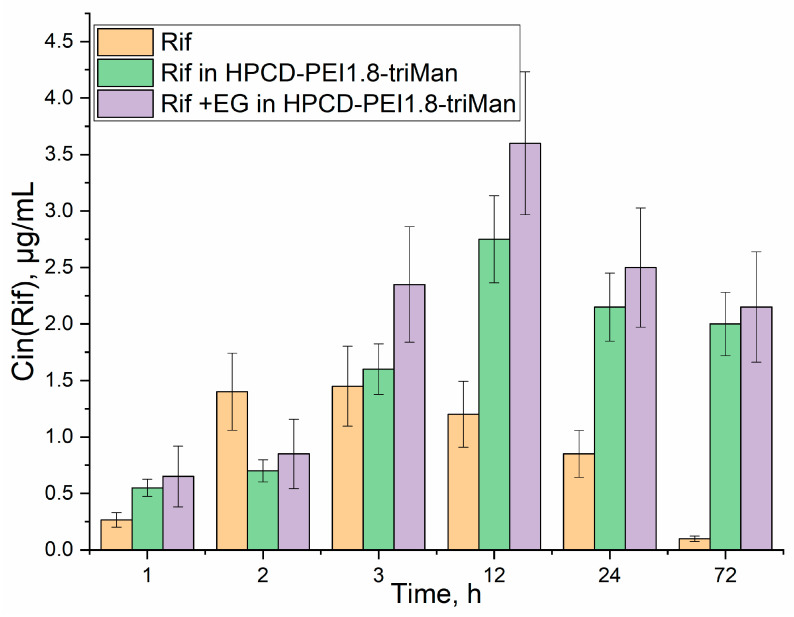
Intracellular (*E. coli*) concentrations of Rif pre-incubated with cells in the form of various formulations: free form, in the delivery system, and enhanced with the adjuvant (EG).

**Figure 4 diagnostics-13-00698-f004:**
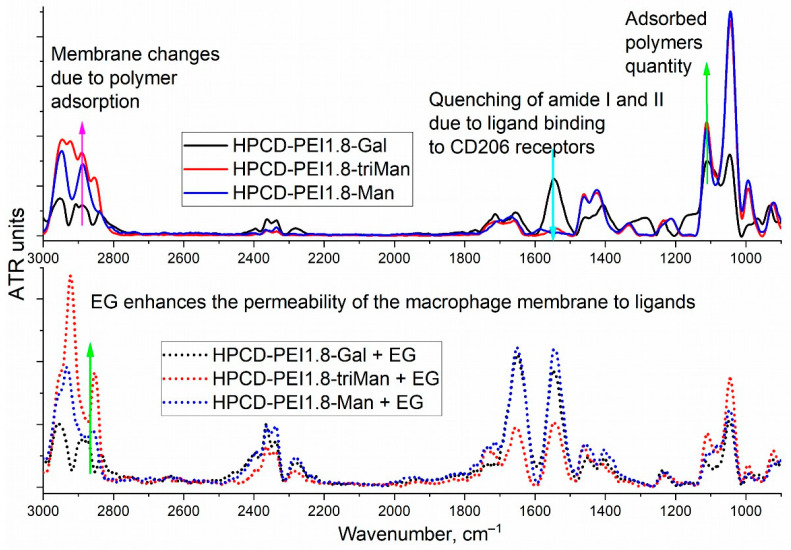
FTIR spectra of CD206+ Macrophages with absorbed E. coli incubated with drug delivery systems with different carbohydrate vectors and eugenol-enhanced formulations.

**Figure 5 diagnostics-13-00698-f005:**
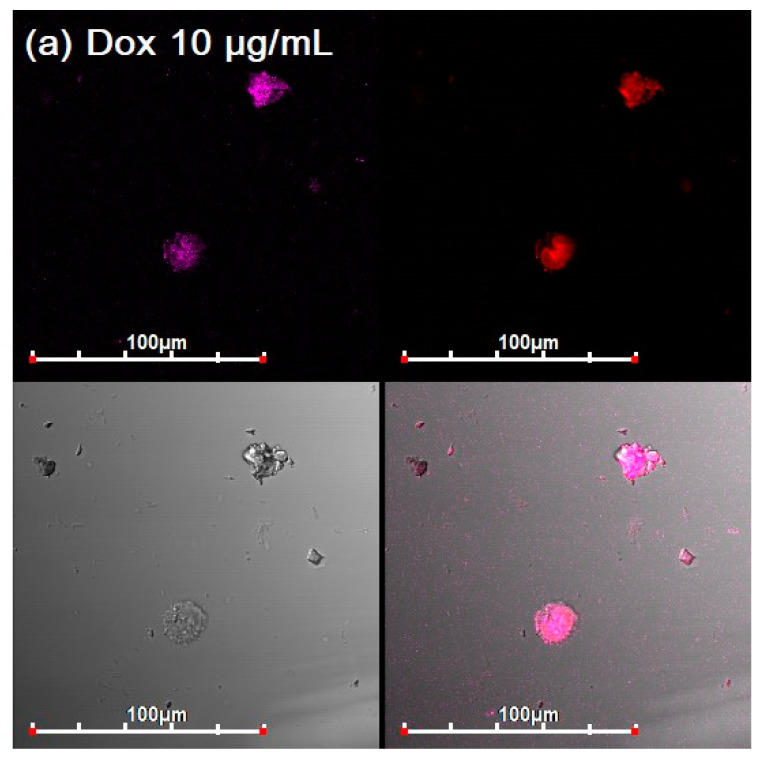
Confocal laser scanning images of CD206+ macrophages with absorbed eosin-labelled *E. coli*. Incubation 2 h with Dox 10 μg/mL and FITC-labeled HPCD-PEI1.8. Dox 10 μg/mL: (**a**) free, (**b**) with 1 mg/mL EG, (**c**) in FITC-labelled HPCD-PEI1.8-Gal, (**d**) in FITC-labelled HPCD-PEI1.8-triMan, (**e**) in FITC-labelled HPCD-PEI1.8-Man, (**f**) in FITC-labelled HPCD-PEI1.8-triMan with 1 mg/mL EG. The scale segment is 100 µm (division value is 20 µm); 4–6 channels are shown: red, Dox; green, FITC; magenta, eosin; gray, transmission light mode; and overlay. λ_em_ = 488 nm (multiline Argon laser).

**Figure 6 diagnostics-13-00698-f006:**
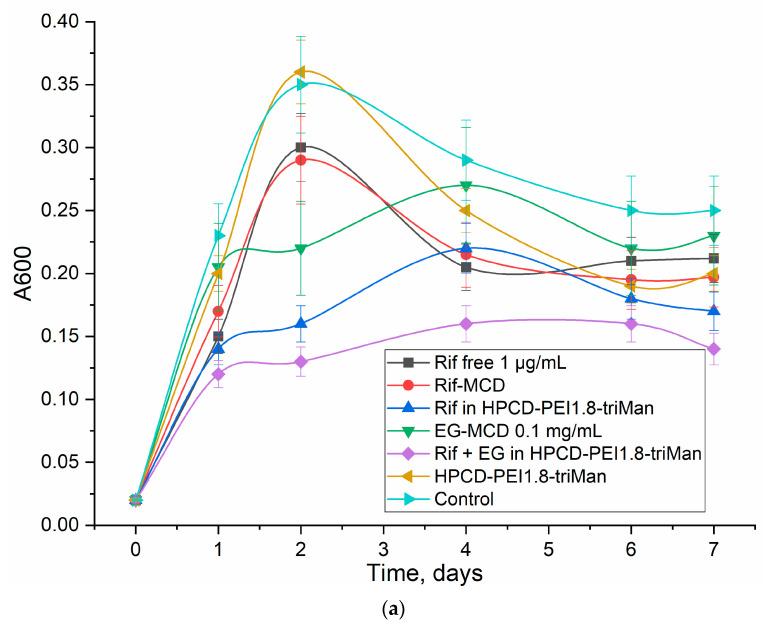
Antibacterial activity of Rif (1 μg/mL) in free form and loaded into HPCD-PEI1.8-triMan, enhanced with eugenol (0.1 mg/mL): (**a**) absorbance of samples (in terms of the diluted samples (absorption was determined in aliquots diluted 4–10 times)) correlated with CFU, and (**b**) CFU dependences determined by quantitative seeding on petri dishes. Rif: polymer mass ratio = 1:10. pH 7.4 (0.01 M PBS), 37 °C.

**Table 1 diagnostics-13-00698-t001:** Characteristics of drug delivery systems and their affinity to mannose receptor.

Carrier *	Label (n) **	Molecular Weight, kDa	Hydrodynamic Size ***, nm	Zeta-Potential ****, mV	Polydispersity Index
HPCD-PEI1.8-FITC * (3:1: 0.5:n) **	Man (12)	9 ± 3	176 ± 100	–4 ± 2	0.4
	triMan (3)		120 ± 50	–6.5 ± 1.5	0.45
	Gal (12)		170 ± 60	–7 ± 3	0.4

* FITC-labeled ligands were used only for experiments with macrophages. ** n is the number of carbohydrate labels. *** by NTA. **** by DLS.

**Table 2 diagnostics-13-00698-t002:** The amounts of Dox and FITC-labeled carrier absorbed by macrophages with E. coli inside. Fluorescence detection (Method section). T = 22 °C.

Sample			Dox Absorbed, %	FITC-Labeled Carrier Absorbed, %
Dox 10 μg/mL	-	-	23 ± 2	-
		Eugenol 1 mg/mL	30 ± 3	
	HPCD-PEI1.8-Gal		27 ± 4	13 ± 4
	HPCD-PEI1.8-triMan		65 ± 4	73 ± 7
	HPCD-PEI1.8-Man		49 ± 3	53 ± 3
	HPCD-PEI1.8-Gal	-	27 ± 5	12 ± 1
	HPCD-PEI1.8-triMan		77 ± 3	63 ± 4
	HPCD-PEI1.8-Man		53 ± 2	57 ± 2

## Data Availability

The data presented in this study are available in the main text and Appendix A.

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
