# Peer review of "Application Prospects of FTIR Spectroscopy and CLSM to Monitor the Drugs Interaction with Bacteria Cells Localized in Macrophages for Diagnosis and Treatment Control of Respiratory Diseases"

_diagnostics, 2023, doi:10.3390/diagnostics13040698_

Round 1
Reviewer 1 Report
The manuscript describes the application of FTIR spectroscopy and CLSM to monitor the interaction of the drug with bacteria cells localized in macrophages for diagnosis and treatment control of respiratory diseases with the aim of development a new approach based on the spectral and optical method for monitoring the course of treatment, potential diagnosis of bacterial diseases and strengthening existing therapeutic formulations. Overall, the methodology of the study is robust, and the results are clearly explained.
Specific comments
Page 4 line 150 – How was the cell separated?
Page 4 line 174 – Were the cells cultured in T75 or T25 flask or plate? What was the volume of the medium added into the flask/plate?
Page 4 line 174 - The medium was replaced with what? Or was it replenished?
Page 5 line 225 – Do you mean three replicates (triplicates)?
Page 7 line 266 – What is the P-value?
Reviewer 2 Report
The manuscript: Application of FTIR spectroscopy and CLSM to monitor the drugs interaction with bacteria cells localized in macrophages for diagnosis and treatment control of respiratory diseases by Elena Vadimovna Kudryashova * and collaborators is a very important contribution to the field of infections and should find interest to scientists from different areas of research. The manuscript describes in details the approach the authors used.
My recommendation is to accept the manuscript for publication.

Author Response
Dear reviewers and editors! The authors of the presented work sincerely thank you for the attentive reading of the article and writing a constructive review! All comments are taken into account. Below is a description of the changes in the work that we made during the revision.
Sincerely yours,
Prof. Dr. Sci. Elena V. Kudryashova,
Chemical enzymology department
Lomonosov Moscow State University
Leninskie gory 1 Moscow Russia 119991
Tel.: +7 495 939 3434
E-mail: helena_koudriachova@hotmail.com
Reviewer 3 Report
In their manuscript entitled “Application of FTIR spectroscopy and CLSM to monitor the drugs interaction with bacteria cells localized in macrophages for diagnosis and treatment control of respiratory diseases”, Igor D. Zlotnikov et al. attempted to demonstrate the usefulness of spectral and optical methods to monitor the infection of macrophages by E. coli. However, in my opinion, promises are not fulfilled. The main problem is that this paper is not focused on a sufficiently well-defined topic and question. At least some experiments were not conducted following an adequate experimental procedure; some important controls are absent or only partly included and the data provided are not sufficient and convincing to me. Authors used their results to provide interpretations that are usually not possible to be done; in other words results are overstated in many cases. As for the manuscript, it is badly written, many explanations are missing (with undefined abbreviations in the text e.g. FTIR, NTA, Cin) and it is difficult to follow the overall rationale of the study. Figures are not well-presented, lacking also explanations. The Introduction is confusing, mixing contextual elements with discussion and opinion from the Authors. In Section 2.8 “Antibacterial activity of Rif”, it is stated “The dependences of CFU (cell viability) on the concentration of Rif were determined by A600.” Then lines 427-428: “Figure 6 shows the curves of the survival of E. coli bacterial cells on the incubation time with various forms of Rif.” However, OD measurements do not allow to determine CFU and cell viability. There is also an obvious lack of rigor; for example, it is stated lines 269-270 “the Rif penetrates into cells after 2–3 h (Figure 3).” However, no results are provided for the 3h time point. Similarly, authors state line 308 “maintained at > 2 μg/mL for 72 h”, although the last time point investigated was 24h. Although there is a section about “Statistical analysis” in Material and Methods (Section 2.9), no statistical comparisons were performed; the term “significant” is used five times in the manuscript, but it is never supported by any p value. This non-exhaustive list of issues leads to the conclusion that this paper is far from being publishable; accordingly, I recommend rejection.
Round 2
Reviewer 3 Report
Igor D. Zlotnikov et al. took into account all the comments received to provide an improved version of their original manuscript. I noted the changes and appreciated the efforts done by the Authors. Important explanations are now provided, which ease the reading and understanding of the whole paper. Some additional improvements may be done, to correct some imprecisions/possible confusions (e.g. it is mentioned in section 2.8 line 252 “seven days”, although Figure 6 reports results up to 6 days), add some further explanations while simplify the writing/style, and edit the English language. Accordingly, I think this paper could be accepted for publication in Diagnostics after minor revision noted.
Author Response
The authors are very grateful to you for your positive feedback on the article. Thank you for appreciating the methods being developed. Below are the comments.
With respect,
Zlotnikov I. et al
Reviewer's general impression:
Igor D. Zlotnikov et al. took into account all the comments received to provide an improved version of their original manuscript. I noted the changes and appreciated the efforts done by the Authors. Important explanations are now provided, which ease the reading and understanding of the whole paper. Some additional improvements may be done, to correct some imprecisions/possible confusions (e.g. it is mentioned in section 2.8 line 252 “seven days”, although Figure 6 reports results up to 6 days), add some further explanations while simplify the writing/style, and edit the English language. Accordingly, I think this paper could be accepted for publication in Diagnostics after minor revision noted.
Answer:
The authors added points corresponding to the seventh day of the experiment with cells to Figure 6a, which serves as a good illustration to the text of the manuscript. Corrections of wording inaccuracies have been made. The English language has been edited.